# Leveraging HIV Care Infrastructures for Integrated Chronic Disease and Pandemic Management in Sub-Saharan Africa

**DOI:** 10.3390/ijerph182010751

**Published:** 2021-10-13

**Authors:** Marie A. Brault, Sten H. Vermund, Muktar H. Aliyu, Saad B. Omer, Dave Clark, Donna Spiegelman

**Affiliations:** 1Center for Interdisciplinary Research in AIDS, Department of Social Behavioral Sciences, Yale School of Public Health, Yale University, New Haven, CT 06510, USA; marie.brault@yale.edu; 2Center for Interdisciplinary Research in AIDS, Department of Epidemiology of Microbial Diseases, Yale School of Public Health, Yale University, New Haven, CT 06510, USA; 3Department of Health Policy, Vanderbilt Institute for Global Health, Vanderbilt University School of Medicine, Nashville, TN 37232, USA; muktar.aliyu@vumc.org; 4Department of Epidemiology of Microbial Diseases, Yale Institute for Global Health, Yale School of Public Health, Yale University, New Haven, CT 06510, USA; saad.omer@yale.edu; 5The Aurum Institute, Johannesburg 2194, South Africa; dave.clark@auruminstitute.org; 6Center for Interdisciplinary Research in AIDS, Department of Biostatistics, Yale School of Public Health, Yale University, New Haven, CT 06510, USA; donna.spiegelman@yale.edu

**Keywords:** non-communicable diseases, HIV, infectious diseases, integrated care, health system strengthening, Sub-Saharan Africa

## Abstract

In Sub-Saharan Africa, communicable and other tropical infectious diseases remain major challenges apart from the continuing HIV/AIDS epidemic. Recognition and prevalence of non-communicable diseases have risen throughout Africa, and the reimagining of healthcare delivery is needed to support communities coping with not only with HIV, tuberculosis, and COVID-19, but also cancer, cardiovascular disease, diabetes, and depression. Many non-communicable diseases can be prevented or treated with low-cost interventions, yet implementation of such care has been limited in the region. In this Perspective piece, we argue that deployment of an integrated service delivery model is an urgent next step, propose a South African model for integration, and conclude with recommendations for next steps in research and implementation. An approach that is inspired by South African experience would build on existing HIV-focused infrastructure that has been developed by Ministries of Health with strong support from the U.S. President’s Emergency Response for AIDS Relief (PEPFAR) and the Global Fund to Fight AIDS, Tuberculosis and Malaria. An integrated chronic healthcare model holds promise to sustainably deliver infectious disease and non-communicable disease care. Integrated care will be especially critical as health systems seek to cope with the unprecedented challenges associated with COVID-19 and future pandemic threats.

## 1. Introduction

In Sub-Saharan Africa (SSA), declining rates of extreme poverty [1,2] have been accompanied by increasing rates of non-communicable diseases (NCDs) [3,4,5,6,7], persistent burdens of infectious disease [8,9,10,11,12], and emerging pathogens such as the novel coronavirus (COVID-19) [13]. Given the magnitude of the epidemics of human immunodeficiency virus/acquired immunodeficiency syndrome (HIV/AIDS) and tuberculosis (TB), targeted resources have come from donor nations, the U.S. President’s Emergency Plan for AIDS Relief (PEPFAR), the Global Fund to Fight AIDS, Tuberculosis and Malaria (Global Fund), and ministries of health to scale-up testing and treatment [14]. This investment has resulted in the dramatic 50% reduction in AIDS-related deaths since 2010 in the Eastern and Southern Africa regions [8], but without commensurate NCD screening and management, there is now considerable inequity in the care of chronic diseases [15,16] COVID-19 has only deepened pre-existing inequities as exemplified by slow vaccine roll-out in SSA [17,18]. Patients with COVID-19 are entering the standard health care system with modestly, at best, enhanced pandemic resources [19].

As countries cope with this historic pandemic and strive for Universal Health Coverage in the Sustainable Development Goal era, new solutions are needed to strengthen health systems, reduce inequities and siloes in service delivery, and meet the challenges posed by emerging pathogens [20]. In this brief Perspective piece, we propose renewed investment in integrated primary care to meet these needs, building on the successes of HIV programming and the Ideal Clinic model in South Africa.

The burden of NCDs in SSA is growing. Of the total burden of disease across SSA from 1990–2017, the proportion of NCDs, increased from 18.6% to 29.8% [21]. Between 10–20 million people in SSA have hypertension; if 70% were treated, 4.5 million deaths could be averted (11.5% of global delayed deaths) [22]. After cardiovascular diseases (contributing 22.9 million Disability Adjusted Life Years (DALYs)), cancers (16.9 million DALYs) and mental disorders (13.6 million DALYs) are the next most significant contributors to the burden of NCDs in SSA [21]. A systematic review finds that population prevalence of Type 2 diabetes varies considerably across region and country, but is as high as 12% in some urban settings [23]. Morbidity and mortality due to NCDs can be prevented with known low-cost, high-impact interventions, and the World Health Organization (WHO) has developed guidelines for packages of care, such as NCD control, maternal and child health, and sexual and reproductive health and rights [24] WHO has described some of these NCD control measures as “best buys”, which are clinically and cost-effective and feasible [24,25] However, national guidelines too often focus on a single disease or issue, inadvertently fostering redundant infrastructures and inefficiencies. Thus, new collaborations and capacity-building will be necessary to expand existing clinical infrastructures to meet the needs of the growing NCD epidemic, which may worsen post-COVID [26] In many venues, particularly in rural SSA, we believe that integrated service delivery that incorporates NCD control into existing HIV and TB testing and treatment programs will be the key to addressing the growing NCD epidemic in SSA. Such integration will also serve as countries rebuild from COVID-19 and look ahead to future pandemic preparedness, by reducing siloed care for HIV/AIDS without consideration of competing causes of death [27].

Throughout SSA, PEPFAR and the Global Fund have partnered with Ministries of Health to establish clinics that have been effective based on multiple metrics. This PEPFAR clinical support era began in 2003–2004 and continues to support increased HIV testing and ART-based clinical care for persons living with HIV (PLHIV) and radical reductions in mother-to-child HIV transmission [28]. The scale-up of PEPFAR-funded programs helped provide ART to more than 15 million people, train over 280,000 new health workers and save over 18 million lives through 2019 [29]. From the perspective of infrastructure and sustainability, PEPFAR has also contributed to effective public-private partnerships to improve medicine supply chains and lab systems [30], as well as capacity building in developing and expanding community health worker cadres, adherence to clinical guidelines and monitoring and evaluation systems [31]. PEPFAR-funded programming has also developed key platforms for community engagement and mobilization.

Too many of these programs, however, remain obligated by funding requirements to focus on HIV and TB without deploying their logistics, facilities, and trained staff for the broader needs expressed by communities. COVID-19 has provided an opportunity to leverage the platforms designed for chronic HIV and TB care to promote COVID-related prevention (testing and contact tracing, vaccination), and encourage patients to return to primary care as COVID-related lockdowns recede [32]. Now is the time to invest in broadening their remits even further into NCDs, endemic infections, and tropical diseases, among others. Environmental threats related to increased vector capacities from climate warming, changing flood-drought cycles, changing air moisture capacities, and climate refugees will increase the urgency of this broader mandate for mitigation of health threats.

Despite PEPFAR’s contributions to intervention development and implementation, much of this work was tightly focused on HIV/TB and not broader health systems strengthening. As PEPFAR funding and goals transition away from a vertical emergency response towards country ownership and sustainability [33,34], we believe it essential that this HIV infrastructure be integrated with other health services to meet broader community needs. Treating a broader array of medical conditions will maximize the political and community support necessary to maintain HIV/TB services on the inevitable day that the U.S. Congress diminishes its PEPFAR support. As the U.S. and other donor countries confront the financial crisis sparked by the COVID-19 pandemic, these funding cuts may arrive sooner than expected.

In this Perspective piece, we provide a brief review of integrated communicable/noncommunicable disease prevention and care models and argue that South Africa’s IDEAL Clinics may offer insights for other countries seeking to extend their HIV infrastructure to address growing NCD and emerging health challenges. We conclude with recommendations for future research and implementation.

## 2. Integrated Service Delivery

There is a growing literature describing different models of successful integrated service delivery in low- and middle-income countries (LMICs) [35,36,37,38,39], particularly for older adults [16,40,41], that could help inform the process of transitioning clinics to broader service delivery. Most of these models integrate clinical services at the point of care. However, more recent recommendations encourage further integration beyond the facility due to the common needs across levels of healthcare systems (Figure 1) [42]. Figure 1 presents common healthcare delivery needs regardless of disease type, at each level of a simplified healthcare system, from the community to the healthcare facility to the national level. Individuals, regardless of diagnosis or status, require engagement at the community level for health prevention, referral, or support. Within clinic or hospital facilities, infrastructure and clinical capacity can be used by patients, regardless of their diagnosis. At the system or national level, leadership and governance (including funding, use of data for decision-making, guideline development and implementation, and logistics and supply chain management) should not be siloed by disease.

Feasibility studies in a variety of low- and middle-income countries have shown promise in integrating HIV care with TB, sexual and reproductive services, primary care services, and NCD screening and treatment services [35,36,37,38,39,43,44]. Integrating infectious and NCD services allows more individuals to be screened, treated, and retained in care for a broader range of conditions [35,36,38,43,44]. Management of HIV-exposed, uninfected children in SSA provides further examples of how integrated clinical management can be implemented, by building on HIV infrastructure. In the era of prevention of mother-to-child transmission, HIV-exposed, uninfected children remain a vulnerable group, as they are more likely to suffer from malnutrition, stunting, anemia, neurologic issues, more severe responses to common respiratory illnesses, and have higher rates of mortality even when they remain uninfected [45,46,47,48,49,50,51,52]. Cohort studies of HIV-exposed, uninfected children and pilot interventions suggest that integrated care systems that include longitudinal follow-up and monitoring, NCD screening and support (nutritional interventions, cognitive and developmental screening and support, maternal health support), and infection prevention and treatment may mitigate the negative impacts of HIV exposure [45,47,53,54,55,56,57,58,59,60,61].

Sustainability is supported using pre-existing infrastructure and resources. Recent costing studies of integrated care in SSA also supports the sustainability of these models. A pilot study in Tanzania and Uganda found that managing two or more conditions in a single participant was cheaper (for both the health system and the patient) than managing multiple conditions separately [62]. A study in Malawi similarly found lower out-of-pocket costs for patients in integrated care, compared to non-integrated care [63]. Intersectoral financing and/or community-based micro-financing to address both multiple determinants of health are also being explored as options to support the scale-up of integrated care [64,65]. Further, many elements of the health system put in place or strengthened for infectious disease prevention and treatment can be applied to NCD care and emerging pathogen needs, or vice versa (Table 1) [42,66]. Table 1 identifies elements of health system building blocks that have been strengthened through HIV capacity-building activities, and how these elements could be applied for communicable (including emerging pathogens) and non-communicable diseases.

Sufficient human resources for health are another key element of sustainable and high-quality integrated care. Task-shifting, or the use of non-physician cadres (nurses, community health workers) to provide care, has been one approach recommended in LMICs to address clinical workforce shortages. Several recent studies have assessed the feasibility, pilot effectiveness, and cost-effectiveness of task-shifting in delivering integrated primary care in SSA, both for PLHIV [56,67,68,69,70,71,72,73,74] and HIV-negative patients [75,76,77,78,79,80]. This growing literature highlights a few key recommendations for implementation. Although there is a need for more effectiveness studies of task-shifting for integrated care, the data to-date suggests that it is feasible, can contribute to improved patient experiences and follow-up, and can be a more cost-effective way to integrate NCD care [68,71,74,75,80,81]. At the same time, task-shifting requires effective facility managers, investment in training for non-physician cadres, and adaptation of the task-shifting approach to the context [56,67,73,75,78,82,83].

Integrated care has long been defensible through a human rights lens--should a PLHIV receive optimized care whilst a person with diabetes cannot access life-saving services [84,85]? This rights-for-all strategy supports underserved persons with accountability for an improved quality of care, as is the case within the PEPFAR stratagem [86].

## 3. A South African Model

We suggest one approach for implementation of such integrated care. South Africa’s IDEAL clinics provide a promising model of integrated HIV/NCD care that could be scaled-up and replicated elsewhere in SSA. The IDEAL Clinic Realisation and Management (ICRM) initiative began in 2013 as part of a series of reforms to implement a National Health Insurance System, expand Universal Health Coverage, and systematically improve South Africa’s primary care clinics [87]. The NCD component of the IDEAL clinic initiative was piloted from 2011–2013 in a sub-set of districts, with promising results, leading to its inclusion in the IDEAL clinic guidelines [88]. However, the concept of integrated primary care as proposed by the ICRM is not new to South Africa. In the mid-20th century, Sidney and Emily Kark and collaborators developed the concept of community-oriented primary care in rural South Africa, integrating both preventive and curative services based on the needs of local communities [89,90]. The IDEAL Clinics represent a modern re-imagining of these concepts [91,92].

A set of tools comprising a manual and an electronic dashboard support the implementation of integrated services and certification as an IDEAL clinic [87]. To receive IDEAL clinic certification, a facility must achieve a metrics score indicating successful implementation of key elements. These elements include good infrastructure; adequate staff; adequate medicine and supplies; strong administrative processes including use of appropriate clinical guidelines and protocols; partner and stakeholder engagement to ensure delivery of high-quality care to the community; and integrated clinical services management that “builds on the strengths of the HIV program to deliver integrated care to patients with chronic and/or acute diseases” [87].

The ICRM process requires that primary care facility managers conduct routine assessments of progress towards certification. Quality improvement plans are then developed to guide further planning and implementation. However, much work is needed, as the implementation of the standards has been slower than planned, and evaluations of the integrated care approach have produced mixed results. As of 2018/2019 (the most recent data available), 55.4% of primary care facilities had achieved IDEAL certification [93]. Two recent studies attribute the slow implementation, in part, to limited governance and leadership that has contributed to poor communication, resource constraints, limited input from frontline managers and staff, and limited accountability [94,95].

Some studies of the IDEAL model have noted that HIV stigma has been reduced by treating both HIV positive and negative patients in the same facilities [96,97,98]. However, implementation across facilities has been uneven, with some facilities better able to implement guidelines with higher fidelity, due primarily to better training and clinical mentorship [99]. Other facilities experience continued deficiencies in facility staffing, physical space, medicines, and equipment that have contributed to limited effectiveness and patient satisfaction [97,100]. Further, studies suggest that although integrated care can be helpful for delivering HIV treatment [101], it is also associated with decreases in new hypertension patients on treatment, due to added burdens placed on a weak primary care system [102]. These mixed findings, combined with a previous review of integration identifying limitations in the data [37,103], suggest that additional work is needed to systematically implement, evaluate, and disseminate integrated care models, such as the IDEAL clinics across SSA. Furthermore, like all systems interventions, they are difficult to sustain since systems regress too readily–but where they are sustained, patients report a much higher level of satisfaction with services received [104].

## 4. Discussion

Despite the challenges associated with shifting to integrated services, the Ugandan and Kenyan SEARCH (Sustainable East Africa Research in Community Health) trial demonstrates how quickly NCD care can be integrated within HIV services with community and health system mobilization [105,106]. In the SEARCH study, HIV treatment as prevention was integrated with diabetes and hypertension screening and management in rural communities with some success in hypertension control [44] However, SEARCH faced challenges in linking and retaining screened patients in care, raising questions regarding the long-term feasibility, scalability, and sustainability of this grant-funded approach, requiring additional implementation studies [105]. Many HIV services in other settings have already made the transition to broad-based NCD care and are actively working to engage the whole patient population in addition to the PLHIV they originally served [107,108]. Throughout the continent, African Ministries of Health have called upon their HIV/AIDS-capacitated laboratories, public health experts, and clinicians to address health service stresses from emerging pandemic threats, whether Ebola virus in West Africa and Congo or COVID-19 in South Africa or Nigeria [19,109,110]. Similarly, NCDs and pandemic preparedness efforts can draw upon the capacity-building accomplished in the HIV/AIDS response.

The time has come to broaden the PEPFAR-developed care capacity to embrace the universal health challenges of modern Africa. Over a decade ago, the U.S. President’s Global Health Initiative in the Barack Obama Administration had a similar vision [111,112] but was never funded substantially by the U.S. Congress. The current COVID-19 pandemic underscores with some urgency the need for the integration of the high-quality HIV treatment and care services (including community-based health workers and contact tracers) with other primary care services, including those concerning COVID-19 [4]. Health systems strengthening that can remedy the challenges facing health care systems throughout Sub-Saharan Africa would enable sustainable capacity for diverse endemic and epidemic threats [4,113,114]. Attempts to broaden this capacity should include input from ministries of health, policy experts, researchers, community-based organizations, and other stakeholders, in addition to considerations of sustainability and robust evaluation of the impact of integration on key HIV, emerging infectious diseases, and NCD indicators.

We believe that the urgency for such transformation of more broadly based primary health care is exacerbated by global warming and climate change that will increasingly strain communities through displaced populations, adverse effects on crops, fish, and grazing animals [115,116], direct temperature and environmental effects on vulnerable populations [117,118,119], expanded vectorial capacities of disease-relevant arthropods and snails [120], and mental health stresses [121]. Health systems must be ready to cope with existing and expanding health challenges, particularly in lower income nations. A recent example is the urgency of SARS-CoV-2 vaccination to prevention COVID-19. The more global transmission, the greater the likelihood of escape mutant RNA viruses such that only a truly global and effective response that subsumes low- and middle-income nations will resolve the pandemic [122,123,124,125]. Fortunately, PEPFAR managers have embraced the need to consider COVID-19 as part of an HIV/AIDS/TB remit [21].

There will be many obstacles for which HIV/AIDS/TB quality improvement and capacity research can inform more integrated programs [48,126,127,128]. Communities and patients themselves are vital contributors to systems improvement through program integrations and community participation [129,130,131]. Despite the challenges associated with integrated care, the IDEAL Clinic and similar approaches offer insights into how existing resources can be leveraged to provide comprehensive primary care in LMICs.

## 5. Conclusions

We believe now is the time for the razor focus of PEPFAR and Global Fund-supported HIV/AIDS/TB clinical services to pivot towards integrated care inclusive of a wider swathe of NCDs and pandemic threats. As we have detailed here, approaches for integrated care exist and offer promise for further adaptation and scale-up throughout Sub-Saharan Africa. Although health system limitations (including human and financial resources) present challenges to the implementation of integrated care, there is growing evidence that integrated care can be more efficient than vertical programming. Broader support from policymakers and communities alike can be expected when integrated services are provided. Broader high-quality service provision may also incentivize healthcare workers to contribute locally rather than migrate abroad [132,133,134]. Future financial assessments can estimate both added medical care costs as well as cost savings from early interventions with primary care and responses to emerging diseases. Further evaluation of integrated care to inform implementation and scale-up can support expanded care for communities dealing with burdens of communicable, non-communicable, and emerging diseases.

## Figures and Tables

**Figure 1 ijerph-18-10751-f001:**
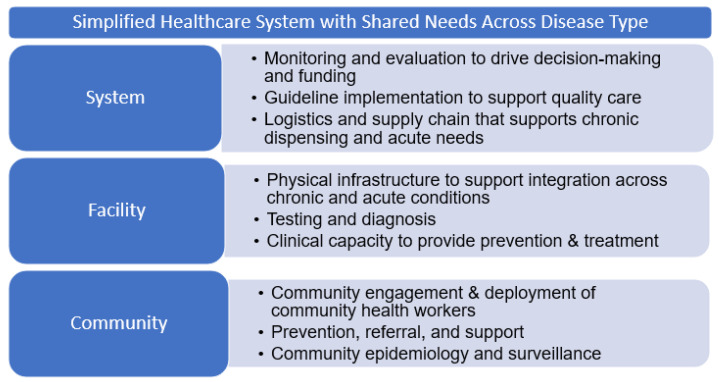
An overview of needs for communicable and non-communicable diseases that could be met with integration across levels of the healthcare system.

**Table 1 ijerph-18-10751-t001:** Examples of shared needs across disease types, and ways in which HIV capacity building could be applied across these needs.

Elements of HIV Capacity-Building	Common Communicable Disease Needs	Common Non-Communicable Disease (NCD) Needs	Emerging and Pandemic Pathogen Needs
Health system strengthening	Expansion of donor-supported health information and monitoring systems beyond HIV/TB Capacity building in health management and quality improvement Systems for immunization, prevention, and treatment supply chains	Data-driven approaches to quality improvement and supply chains Clinic facility improvement to support screening, community services, diagnosis and treatment for simple NCDs Systems for chronic disease medication dispensing	Preparedness for pandemic arrival and outbreaks Planning for healthcare worker safety and infection control Management of testing, vaccine, and treatment supply chains
Clinical capacity for management of complex patients	Prevention through pre-exposure prophylaxis Management of HIV and related opportunistic infections Modify care per immunologic, virologic, and clinical response to antiretroviral therapies and other therapies Retention and follow-up services	Implementation of WHO guidelines and “best buys” for prevention Field management and monitoring (blood sugar, blood pressure) Provision of simple treatments at local facilities through task-shifting Retention and follow-up services	Case recognition and rapid diagnosis to facilitate contact tracing Implementation of infection control guidelines Evidence-based implementation and review of management and treatment guidelines
Laboratories and point-of-care testing, diagnosis, monitoring	Diagnosis of HIV and related diseases Monitoring CD4^+^ cells, virologic, metabolic and hematologic parameters related to common treatments	Monitoring chronic diseases and complications Monitoring impact of common therapies	Diagnosis of emerging and re-emerging pathogens Sequencing and monitoring for variants of concern
Community engagement and referral, local delivery of interventions	Development of community advisory and governance structures Distribution of prevention and treatment tools (e.g., directly observed therapy (short course), condoms, mosquito nets)	Community-based prevention and adherence support Linkages between community and facility for prevention and treatment	Contact tracing and support for quarantine Home-based care for less severe illness Linkage to testing, vaccination, medical treatment

## Data Availability

No new data were created or analyzed in this study. Data sharing is not applicable to this article.

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
