# Peer review of "Leveraging HIV Care Infrastructures for Integrated Chronic Disease and Pandemic Management in Sub-Saharan Africa"

_ijerph, 2021, doi:10.3390/ijerph182010751_

Round 1

Reviewer 1 Report

Thank you for the opportunity to review this thoughtful commentary on the need for integration of preventive and curative care for chronic and infectious conditions in resource limited settings.  Using the example of the South African IDEAL clinic model, the authors present a compelling case for the need for clinical integration, particularly in light of the current COVID-19 pandemic.

My suggestions for the authors are minimal and optional:

  1. While the article focuses on the "why" of integrative health services, the reader would also benefit from more input on the "how."  As the authors point out with both the IDEAL clinic and SEARCH study examples, implementation of integrated services is possible, but the sustainability and long-term impact have yet to be fully demonstrated.  One factor limiting this impact to date is serious workforce challenges, ranging from poor provider to population ratios, shortages of higher skilled providers (RN, NP, PA, MD), and regulatory barriers to task shifting. This paper would benefit from an additional paragraph or so on task shifting, training, provider pipeline and other workforce issues, perhaps considering data from Seidman and Atun (doi: 10.1186/s12960-017-0200-9).
  2. More emphasis on the “how” could also be presented in a review of examples of routine clinical integration from the management of HIV exposed uninfected children in Sub-Saharan Africa, such as in Zambia and South Africa. Data from the IAS 2019 HIV-Exposed Uninfected Child and Adolescent Workshop may be particularly relevant for the current paper.
  3. The authors final points regarding expansion of the increasing laser focus of PEPFAR and the Global Fund support are critically important. These points could be underscored further with data from models and projections of the impact of chronic and communicable disease in Sub-Saharan Africa and the current overall COVID vaccine coverage estimates of <2% to date (https://africacdc.org/covid-19-vaccination/).
  4. Minor copyediting suggestions include:
  • Line 39, separate HIV/TB to not imply coinfection mortality only
  • Line 68, add PEPFAR clinical support era
  • Lines 78-82, consider reframing as programs obligated to focus on HIV and TB to comply with funding requirements; consider reframing COVID experience as an opportunity from which to build integrated service delivery implementation
  • Table 1, consider adding expansion of donor supported health information and monitoring systems to include chronic diseases and other communicable diseases; consider adding screening and community services to clinic facility improvement; consider adding retention services to clinical capacity; consider adding virologic monitoring
  • Line 131, correct ICRM to Ideal Clinic Realisation and Maintenance

In summary, I highly recommend this article for publication in the International Journal of Environmental Research and Public Health, with only minor optional suggestions for improvement.

Author Response

Reviewer 1

Thank you for the opportunity to review this thoughtful commentary on the need for integration of preventive and curative care for chronic and infectious conditions in resource limited settings.  Using the example of the South African IDEAL clinic model, the authors present a compelling case for the need for clinical integration, particularly in light of the current COVID-19 pandemic.

My suggestions for the authors are minimal and optional:

  1. While the article focuses on the "why" of integrative health services, the reader would also benefit from more input on the "how."  As the authors point out with both the IDEAL clinic and SEARCH study examples, implementation of integrated services is possible, but the sustainability and long-term impact have yet to be fully demonstrated.  One factor limiting this impact to date is serious workforce challenges, ranging from poor provider to population ratios, shortages of higher skilled providers (RN, NP, PA, MD), and regulatory barriers to task shifting. This paper would benefit from an additional paragraph or so on task shifting, training, provider pipeline and other workforce issues, perhaps considering data from Seidman and Atun (doi: 10.1186/s12960-017-0200-9).

Response: Thank you for the suggestion, and we agree that the role of task-shifting in the implementation of integrated primary care is important, for both PLHIV and HIV-negative patients. We have added a paragraph on page 4 that briefly describes positive impacts of task-shifting and requirements for effective task-shifting within integrated primary care settings in Sub-Saharan Africa. 

  1. More emphasis on the “how” could also be presented in a review of examples of routine clinical integration from the management of HIV exposed uninfected children in Sub-Saharan Africa, such as in Zambia and South Africa. Data from the IAS 2019 HIV-Exposed Uninfected Child and Adolescent Workshop may be particularly relevant for the current paper.

Response: We thank the reviewer for pointing us to this workshop and body of literature. We have added a section briefly reviewing examples of integrated care for HIV-exposed uninfected children on page 3, in the “Integrated Service Delivery” section.  

  1. The authors final points regarding expansion of the increasing laser focus of PEPFAR and the Global Fund support are critically important. These points could be underscored further with data from models and projections of the impact of chronic and communicable disease in Sub-Saharan Africa and the current overall COVID vaccine coverage estimates of <2% to date (https://africacdc.org/covid-19-vaccination/).

Response: We have added additional references and data concerning the magnitude of the NCD challenges.

  1. Minor copyediting suggestions include:
  • Line 39, separate HIV/TB to not imply coinfection mortality only

Response: We have removed TB to simplify and clarify the reduction in HIV mortality in the PEPFAR era, adding numerical data and a reference to support this, per Reviewer 4’s comment.

  • Line 68, add PEPFAR clinical support era

Response: We have made this correction.

  • Lines 78-82, consider reframing as programs obligated to focus on HIV and TB to comply with funding requirements; consider reframing COVID experience as an opportunity from which to build integrated service delivery implementation

Response: We have made these edits.

  • Table 1, consider adding expansion of donor supported health information and monitoring systems to include chronic diseases and other communicable diseases; consider adding screening and community services to clinic facility improvement; consider adding retention services to clinical capacity; consider adding virologic monitoring

Response: Thank you for the suggestions. These edits have been made.

  • Line 131, correct ICRM to Ideal Clinic Realisation and Maintenance

Response: We have made this correction.

In summary, I highly recommend this article for publication in the International Journal of Environmental Research and Public Health, with only minor optional suggestions for improvement.

Reviewer 2 Report

Comments on Leveraging HIV Care Infrastructures for Integrated Chronic 2 Disease and Pandemic Management in Sub-Saharan Africa 

General Comments: 

  1. This paper is a well researched and well presented piece of work.  The authors have made a clear and convincing arguments as to the need to develop integrated programs to manage related chronic diseases.  The references present a review of updated literature on the topic.
  2. I have one major concern that I believe needs to be addressed before publication. The question is about the finances for the integrated model. The authors note foreign funding has been decreasing particularly in the age of COVID-19. What are the financial implications for changes ? Without addressing this key issue, the research does not provide a sound basis for the required changes for health care delivery. It leaves the reader with questions about change and sustainability.
  3. I suggest the authors provide a section on the financial implications of creating integrated management.
  4. Although health system challenges to this integration have been discussed it would strengthen the argument to suggest ways of overcoming these barriers.
  1. Specific Comments:
  1. Please check the manuscript to make sure that the acronyms have been spelled out in full.

Author Response

Reviewer 2

Comments on Leveraging HIV Care Infrastructures for Integrated Chronic 2 Disease and Pandemic Management in Sub-Saharan Africa 

General Comments: This paper is a well researched and well presented piece of work.  The authors have made a clear and convincing arguments as to the need to develop integrated programs to manage related chronic diseases.  The references present a review of updated literature on the topic.

  1. I have one major concern that I believe needs to be addressed before publication. The question is about the finances for the integrated model. The authors note foreign funding has been decreasing particularly in the age of COVID-19. What are the financial implications for changes ? Without addressing this key issue, the research does not provide a sound basis for the required changes for health care delivery. It leaves the reader with questions about change and sustainability. I suggest the authors provide a section on the financial implications of creating integrated management.

Response:  We understand and sympathize with the reviewer’s concern, but a financial assessment is far beyond the scope of our commentary.  We believe that the commentary will stimulate such an assessment, but will not be within our remit for this relatively short review.  To acknowledge this limitation, we have added the following sentence to our conclusion, “Future financial assessments can estimate both added medical care costs as well as cost savings from early interventions with primary care and responses to emerging diseases.”

  1. Although health system challenges to this integration have been discussed it would strengthen the argument to suggest ways of overcoming these barriers.

Response: In response to Reviewer 1’s recommendation to add more on the “how” of overcoming barriers to implementing integrated care, as well as Reviewer 2’s recommendation to address approaches for funding integrated care, we have added additional sections on task-shifting, clinical integration for HIV exposed youth, and financing approaches.

Specific Comments:

  1. Please check the manuscript to make sure that the acronyms have been spelled out in full.

Response: We have double-checked the manuscript, and added the meaning of the SEARCH study acronym (Sustainable East Africa Research in Community Health).

Reviewer 3 Report

The focus of this paper is interesting for the scientific community and the public in general, since it analysis integrated care to cope with COVID-19 and future pandemic threats. This manuscript proposes an integrated chronic healthcare model with the aim to deliver infectious disease and non-communicable disease care in a sustainable manner. The authors base their model in the South African experience, it integrates HIV-focused infrastructures that have been developed by Ministries of Health with strong support from the U.S. Presidents Emergency Response for AIDS Relief (PEPFAR) and the Global Fund to Fight AIDS, Tuberculosis and Malaria.

Comments and suggestions:

-

  • The introduction needs a last paragragh explaining the structure of the paper.
  • Figure 1 needs to be better explained in the text. What do you mean by system, facility and community? These are very broad terms that need to be specified in you applied study.
  • Table 1 need to be restructured, it is very messy and not well designed. We get lost reading it. Again it needs to be explained in the text.
  • The section entitled “urgent needs” seems to be a discussion section. Please consider the title “discussion”.
  • This manuscript needs a conclusion section.

Author Response

Reviewer 3

The focus of this paper is interesting for the scientific community and the public in general, since it analysis integrated care to cope with COVID-19 and future pandemic threats. This manuscript proposes an integrated chronic healthcare model with the aim to deliver infectious disease and non-communicable disease care in a sustainable manner. The authors base their model in the South African experience, it integrates HIV-focused infrastructures that have been developed by Ministries of Health with strong support from the U.S. Presidents Emergency Response for AIDS Relief (PEPFAR) and the Global Fund to Fight AIDS, Tuberculosis and Malaria.

Comments and suggestions:

  • The introduction needs a last paragraph explaining the structure of the paper.

Response: We have added a short paragraph to the end of the introduction describing the objectives and structure of the paper. This paragraph reads as follows: “In this “Short Communication” we provide a brief review of integrated communicable/noncommunicable disease prevention and care models, and argue that South Africa’s IDEAL Clinics may offer insights for other countries seeking to extend their HIV infrastructure to address growing NCD and emerging health challenges. We conclude with recommendations for future research and implementation.”

  • Figure 1 needs to be better explained in the text. What do you mean by system, facility and community? These are very broad terms that need to be specified in you applied study.

Response: We have clarified that Figure 1 represents a simplified healthcare system, consisting of the community, facility, and national/system levels. At each level, there are needs shared regardless of diagnosis or disease type. We have added the following text to further explain Figure 1, right after we introduced it in the text: “Figure 1 presents common healthcare delivery needs regardless of disease type, at each level of a simplified healthcare system, from the community to the healthcare facility to the national level. Individuals, regardless of diagnosis or status, require engagement at the community level for health prevention, referral, or support. Within clinic or hospital facilities, infrastructure and clinical capacity can be used by patients, regardless of their diagnosis. At the system or national level, leadership and governance (including funding, use of data for decision-making, guideline development and implementation, and logistics and supply chain management) should not be siloed by disease.”

  • Table 1 need to be restructured, it is very messy and not well designed. We get lost reading it. Again it needs to be explained in the text.

Response: We have edited the wording within Table 1, per Reviewer 1’s suggestions, and we hope that this has improved the readability of the table. We have also added the following sentence after we introduce the table in the text (added text italicized), Table 1 identifies elements of health system building blocks that have been strengthened through HIV capacity-building activities, and how these elements could be applied for communicable, non-communicable, and emerging pathogen diseases.”

  • The section entitled “urgent needs” seems to be a discussion section. Please consider the title “discussion”.

Response: We have changed the title of this section to “Discussion”.

  • This manuscript needs a conclusion section.

Response: We have added a conclusion section summarizing the main points of the piece.

Reviewer 4 Report

Thank you for the opportunity to review your paper about leveraging HIV care infrastructures for integrated chronic disease and pandemic management in Sub-Saharan Africa.

The abstract includes the main information about the study. However, I suggest that the authors add information about the objectives of the paper.

The introduction of the paper must be clearly identified.

On page 1 the authors refer: “This investment has resulted in the dramatic reduction in HIV/TB mortality (…)”. I think it is important to present numerical results that support this statement.

It would be important to describe the epidemiological situation using incidence and prevalence values.

On page 3, I think it is necessary for the authors to better explain the diagram in Figure 1.

I suggest that the authors add a conclusions section, where they summarize the information described, put the difficulties encountered and the implications for practice.

Author Response

Reviewer 4

  • Thank you for the opportunity to review your paper about leveraging HIV care infrastructures for integrated chronic disease and pandemic management in Sub-Saharan Africa.
  • The abstract includes the main information about the study. However, I suggest that the authors add information about the objectives of the paper.

Response: We have revised the abstract to articulate the objectives of the paper. The abstract now reads, “In Sub-Saharan Africa, communicable and other tropical infectious diseases remain major challenges apart from the continuing HIV/AIDS pandemic. Recognition and prevalence of non-communicable diseases have risen throughout Africa, and the reimagining of healthcare delivery is needed to support communities coping with not only HIV, tuberculosis, and COVID-19, but also cancer, cardiovascular disease, diabetes, and depression. Many non-communicable diseases can be prevented or treated with low-cost interventions, yet implementation of such care has been limited in the region. In this Short Communication piece, we argue that deployment of an integrated service delivery model is an urgent next step, present a South African model for integration, and conclude with recommendations for next steps in research and implementation. An approach that is inspired by South African experience would build on existing HIV-focused infrastructures that have been developed by Ministries of Health with strong support from the U.S. President’s Emergency Response for AIDS Relief (PEPFAR) and the Global Fund to Fight AIDS, Tuberculosis and Malaria. An integrated chronic healthcare model holds promise to sustainably deliver infectious disease and non-communicable disease care. Integrated care will be especially critical as health systems seek to cope with the unprecedented challenges associated with COVID-19 and future pandemic threats.”

  • The introduction of the paper must be clearly identified.

Response: We have added an “Introduction” heading to the beginning of the manuscript.

  • On page 1 the authors refer: “This investment has resulted in the dramatic reduction in HIV/TB mortality (…)”. I think it is important to present numerical results that support this statement.

Response: We have revised this sentence to clarify that we are referring to HIV/AIDS-mortality, not mortality due to HIV/TB co-infection. We have also added a reference from the 2021 UNAIDS Global AIDS update, and stated that there has been a 50% reduction in AIDS-related deaths since 2010 in the Eastern and Southern Africa regions. The revised sentence now states (edits italicized): This investment has resulted in the dramatic 50% reduction in AIDS-related deaths since 2010 in the Eastern and Southern Africa regions,(9) but without commensurate NCD screening and management, there is now considerable inequity in the care of chronic diseases.(10,11)

  • It would be important to describe the epidemiological situation using incidence and prevalence values.

Response: The epidemiological situation of NCDs in Sub-Saharan Africa is complex and highly variable, but we have attempted to provide additional data and references on this on page 2.

  • On page 3, I think it is necessary for the authors to better explain the diagram in Figure 1.

Response: Please see our response to Reviewer 3 above. We have added additional text to explain Figure 1.

  • I suggest that the authors add a conclusions section, where they summarize the information described, put the difficulties encountered and the implications for practice.

Response: See our response to reviewer 3 above. We have added a conclusion section.

Round 2

Reviewer 2 Report

I think the article is now ready for publication.  Addressing sustainability rather than specific examination of financial implications answers my concerns about the future of this approach. I did not expect a investigation into exact financial support.  Also the  expanded discussion about overcoming barriers supports the question of how this approach can be maintained.  

This manuscript is a resubmission of an earlier submission. The following is a list of the peer review reports and author responses from that submission.